

# CoproID predicts the source of coprolites and paleofeces using microbiome composition and host DNA content

Maxime Borry[1], Bryan Cordova[1], Angela Perri[2,3], Marsha Wibowo[4,5,6], Tanvi Prasad Honap[7,8], Jada Ko[9], Jie Yu[10], Kate Britton[3,11], Linus Girdland-Flink[11,12], Robert C. Power[3,13], Ingelise Stuijts[14], Domingo C. Salazar-García[15,16], Courtney Hofman[7,8], Richard Hagan[1], Thérèse Samdapawindé Kagoné[17], Nicolas Meda[17], Helene Carabin[18], David Jacobson[7,8], Karl Reinhard[19], Cecil Lewis[7,8], Aleksandar Kostic[4,5,6], Choongwon Jeong[1,20], Alexander Herbig[1], Alexander Hübner[1] and Christina Warinner[1,9,21]

[1] Department of Archaeogenetics, Max Planck Institute for the Science of Human History, Jena, Germany
[2] Department of Archaeology, Durham University, Durham, UK
[3] Department of Human Evolution, Max Planck Institute for Evolutionary Anthropology, Leipzig, Germany
[4] Section on Pathophysiology and Molecular Pharmacology, Joslin Diabetes Center, Boston, MA, USA
[5] Section on Islet Cell and Regenerative Biology, Joslin Diabetes Center, Boston, MA, USA
[6] Department of Microbiology, Harvard Medical School, Boston, MA, USA
[7] Department of Anthropology, University of Oklahoma, Norman, OK, USA
[8] Laboratories of Molecular Anthropology and Microbiome Research (LMAMR), University of Oklahoma, Norman, OK, USA
[9] Department of Anthropology, Harvard University, Cambridge, MA, USA
[10] Department of History, Wuhan University, Wuhan, China
[11] Department of Archaeology, University of Aberdeen, Aberdeen, Scotland, UK
[12] School of Natural Sciences and Psychology, Liverpool John Moores University, Liverpool, UK
[13] Institut für Vor- und Frühgeschichtliche Archäologie und Provinzialrömische Archäologie, Ludwig-Maximilians-Universität München, München, Germany
[14] The Discovery Programme, Dublin, Ireland
[15] Grupo de Investigación en Prehistoria IT-1223-19 (UPV-EHU), IKERBASQUE-Basque Foundation for Science, Vitoria-Gasteiz, Spain
[16] Departament de Prehistòria, Arqueologia i Història Antiga, Universitat de València, València, Spain
[17] Centre Muraz, Bobo-Dioulasso, Burkina Faso
[18] Département de pathologie et de microbiologie, Faculté de Médecine vétérinaire, Université de Montréal, Saint-Hyacinthe, QC, Canada
[19] School of Natural Resources, University of Nebraska, Lincoln, NE, USA
[20] School of Biological Sciences, Seoul National University, Seoul, South Korea
[21] Faculty of Biological Sciences, Friedrich-Schiller Universität Jena, Jena, Germany

Corresponding authors
Maxime Borry, borry@shh.mpg.de
Christina Warinner,
warinner@fas.harvard.edu

## ABSTRACT

Shotgun metagenomics applied to archaeological feces (paleofeces) can bring new insights into the composition and functions of human and animal gut microbiota from the past. However, paleofeces often undergo physical distortions in archaeological sediments, making their source species difficult to identify on the basis of fecal morphology or microscopic features alone. Here we present a reproducible and scalable pipeline using both host and microbial DNA to infer the host source of fecal material. We apply this pipeline to newly sequenced archaeological

specimens and show that we are able to distinguish morphologically similar human and canine paleofeces, as well as non-fecal sediments, from a range of archaeological contexts.

## INTRODUCTION

The gut microbiome, located in the distal colon and primarily studied through the analysis of feces, is the largest and arguably most influential microbial community within the body (*The Human Microbiome Project Consortium, 2012*). Recent investigations of the human microbiome have revealed that it plays diverse roles in health and disease, and gut microbiome composition has been linked to a variety of human health states, including inflammatory bowel diseases, diabetes, and obesity (*Kho & Lal, 2018*). To investigate the gut microbiome, metagenomic sequencing is typically used to reveal both the taxononomic composition (i.e., which bacteria are there) and the functions the microbes are capable of performing (i.e., their potential metabolic activities) (*Sharpton, 2014*). Given the importance of the gut microbiome in human health, there is great interest in understanding its recent evolutionary and ecological history (*Warinner & Lewis, 2015*; *Davenport et al., 2017*).

Paleofeces, either in an organic or partially mineralized (coprolite) state, present a unique opportunity to directly investigate changes in the structure and function of the gut microbiome through time (*Warinner et al., 2015*). Paleofeces are found in a wide variety of archaeological contexts around the world and are generally associated with localized processes of dessication, freezing, or mineralization. Paleofeces can range in size from whole, intact fecal pieces (*Jiménez et al., 2012*) to millimeter-sized sediment inclusions identifiable by their high phosphate and fecal sterol content (*Sistiaga et al., 2014*). Although genetic approaches have long been used to investigate dietary DNA found within human (*Gilbert et al., 2008*; *Poinar et al., 2001*) and animal (*Poinar et al., 1998*; *Hofreiter et al., 2000*; *Bon et al., 2012*; *Wood et al., 2016*) paleofeces, it is only recently that improvements in metagenomic sequencing and bioinformatics have enabled detailed characterization of their microbial communities (*Tito et al., 2008, 2012*; *Warinner et al., 2017*).

However, before evolutionary studies of the gut microbiome can be conducted, it is first necessary to confirm the host source of the paleofeces under study. Feces can be difficult to taxonomically assign by morphology alone (Supplemental Text; *Reinhard & Bryant, 1992*), and human and canine feces can be particularly difficult to distinguish in archaeological contexts (*Poinar et al., 2009*). Since their initial domestication more than 12,000 years ago (*Frantz et al., 2016*), dogs have often lived in close association with humans, and it is not uncommon for human and dog feces to co-occur at archaeological sites. Moreover, dogs often consume diets similar to humans because of provisioning or

refuse scavenging (*Guiry, 2012*), making their feces difficult to distinguish based on dietary contents. Even well-preserved fecal material degrades over time, changing in size, shape, and color (Fig. 1; *Reinhard & Bryant, 1992*). The combined analysis of host and microbial ancient DNA (aDNA) within paleofeces presents a potential solution to this problem.

Previously, paleofeces host source has been genetically inferred on the basis of PCR-amplified mitochondrial DNA sequences alone (*Hofreiter et al., 2000*); however, this is problematic in the case of dogs, which, in addition to being pets and working animals, were also eaten by many ancient cultures (*Clutton-Brock & Hammond, 1994*; *Rosenswig, 2007*; *Kirch & O'Day, 2003*; *Podberscek, 2009*), and thus trace amounts of dog DNA may be expected to be present in the feces of humans consuming dogs. Additionally, dogs often scavenge on human refuse, including human excrement (*Butler & Du Toit, 2002*), and thus ancient dog feces could also contain trace amounts of human DNA, which could be further inflated by PCR-based methods.

A metagenomics approach overcomes these issues by allowing a quantitative assessment of eukaryotic DNA at a genome-wide scale, including the identification and removal of modern human contaminant DNA that could potentially arise during excavation or subsequent curation or storage. It also allows for the microbial composition of the feces to be taken into account. Gut microbiome composition differs among mammal species (*Ley et al., 2008*), and thus paleofeces microbial composition could be used to confirm and authenticate host assignment. Available microbial tools, such as SourceTracker (*Knights et al., 2011*) and FEAST (*Shenhav et al., 2019*), can be used to perform the source prediction of microbiome samples from uncertain sources (sinks) using a reference dataset of source-labeled microbiome samples and, respectively, Gibbs sampling or an Expectation-Maximization algorithm. However, although SourceTracker has been widely used for modern microbiome studies and has even been applied to ancient gut microbiome data (*Tito et al., 2012*; *Hagan et al., 2020*), it was not designed to be a host species identification tool for ancient microbiomes.

In this work we present a bioinformatics method to infer and authenticate the host source of paleofeces from shotgun metagenomic DNA sequencing data: coproID (coprolite IDentification). coproID combines the analysis of putative host ancient DNA with a machine learning prediction of the feces source based on microbiome taxonomic composition. Ultimately, coproID predicts the host source of a paleofeces specimen from the shotgun metagenomic data derived from it. We apply coproID to previously published modern fecal datasets and show that it can be used to reliably predict their host. We then apply coproID to a set of newly sequenced paleofeces specimens and non-fecal archaeological sediments and show that it can discriminate between feces of human and canine origin, as well as between fecal and non-fecal samples.

## MATERIALS AND METHODS

### Gut microbiome reference datasets

Previously published modern reference microbiomes were chosen to represent the diversity of potential paleofeces sources and their possible contaminants, namely human fecal microbiomes from Non-Westernized Human/Rural (NWHR) and Westernized

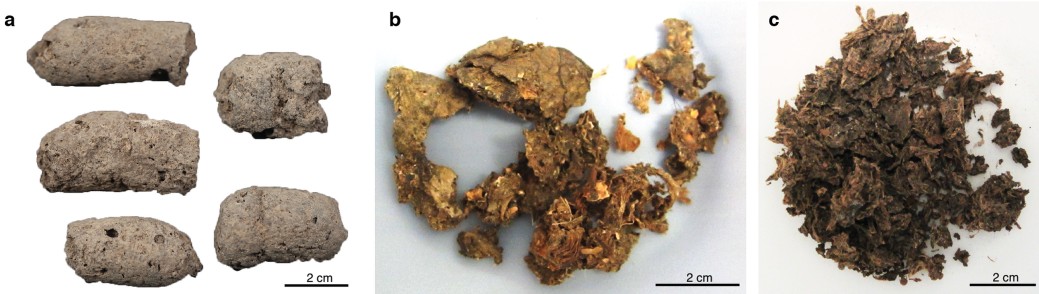

**Figure 1 Examples of archaeological paleofeces analyzed in this study.** (A) H29-3, from Anhui Province, China, Neolithic period; (B) Zape 2, from Durango, Mexico, ca. 1300 BP; (C) Zape 28, from Durango, Mexico, ca. 1300 BP. Paleofeces ranged from slightly mineralized intact pieces (A) to more fragmentary organic states (B and C), and color ranged from pale gray (A) to dark brown (C).

**Table 1 Modern reference microbiome datasets.**

| Metagenome source | Food production | N | Analysis | Source |
|---|---|---|---|---|
| *Homo sapiens*, USA | WHU | 36 | microbiome | *The Human Microbiome Project Consortium (2012)* |
| *Homo sapiens*, India (Bhopal and Kerala) | WHU and NWHR | 19 | microbiome | *Dhakan et al. (2019)* |
| *Homo sapiens*, Fiji (agrarian villages) | NWHR | 20 | microbiome | *Brito et al. (2019)* |
| *Homo sapiens*, Madagascar | NWHR | 110 | microbiome | *Pasolli et al. (2019)* |
| *Homo sapiens*, Brazil (Yanomami) | NWHR | 3 | microbiome | *Pasolli et al. (2019)* |
| *Homo sapiens*, Peru (Tunapuco) | NWHR | 12 | microbiome | *Obregon-Tito et al. (2015)* |
| *Homo sapiens*, Tanzania (Hadza) | NWHR | 38 | microbiome | *Rampelli et al. (2015)* |
| *Homo sapiens*, Peru (Matses) | NWHR | 24 | microbiome | *Obregon-Tito et al. (2015)* |
| *Homo sapiens*, USA (Boston) | WHU | 49 | host DNA | This study |
| *Homo sapiens*, Burkina Faso | NWHR | 69 | host DNA | This study |
| Canis familiaris | – | 150 | microbiome and host DNA | *Coelho et al. (2018)* |
| Soil | – | 16 | microbiome | *Fierer et al. (2012)* |
| Soil | – | 2 | microbiome | *CSIR-Central Institute of Medicinal & Aromatic Plants (2016)* |
| Soil | – | 2 | microbiome | *Orellana et al. (2018)* |

Human/Urban (WHU) communities, dog fecal microbiomes, and soil samples (Table 1). Because the human datasets had been filtered to remove human genetic sequences prior to database deposition, we additionally generated new sequencing data from 118 fecal specimens from both NWHR and WHU populations (Table S5) in order to determine the average proportion and variance of host DNA in human feces. The Joslin Diabetes Center granted Ethical approval (CHS# 2017-25) to sample the WHU individuals. The Centre MURAZ Research Institute granted Ethical approval (No. 31/2016/CE-CM) to sample the NWHR individuals.

![PeerJ]

**Table 2 Archaeological samples.**

| Archeological ID | Laboratory ID | Site Name | Region | Period | Sample type | Archaeologically suspected species | Plot ID |
|---|---|---|---|---|---|---|---|
| Zape 2* | ZSM002 | Cueva de los Muertos Chiquitos | Mexico | 1300 BP | Paleofeces | HUMAN | 01 |
| Zape 5* | ZSM005 | Cueva de los Muertos Chiquitos | Mexico | 1300 BP | Paleofeces | HUMAN | 02 |
| Zape 23 | ZSM023 | Cueva de los Muertos Chiquitos | Mexico | 1300 BP | Paleofeces | HUMAN or CANID | 03 |
| Zape 25 | ZSM025 | Cueva de los Muertos Chiquitos | Mexico | 1300 BP | Paleofeces | HUMAN | 04 |
| Zape 27 | ZSM027 | Cueva de los Muertos Chiquitos | Mexico | 1300 BP | Paleofeces | HUMAN | 05 |
| Zape 28* | ZSM028 | Cueva de los Muertos Chiquitos | Mexico | 1300 BP | Paleofeces | HUMAN | 06 |
| Zape 29 | ZSM029 | Cueva de los Muertos Chiquitos | Mexico | 1300 BP | Paleofeces | HUMAN | 07 |
| Zape 31 | ZSM031 | Cueva de los Muertos Chiquitos | Mexico | 1300 BP | Paleofeces | HUMAN | 08 |
| H29-1 | AHP001 | Xiaosungang | China | Neolithic 7200–6800 BP | Paleofeces | CANID or CERVID | 09 |
| H35-1 | AHP002 | Xiaosungang | China | Neolithic 7200–6800 BP | Paleofeces | CANID or CERVID | 10 |
| H29-2 | AHP003 | Xiaosungang | China | Neolithic 7200–6800 BP | Paleofeces | CANID or CERVID | 11 |
| H29-3 | AHP004 | Xiaosungang | China | Neolithic 7200–6800 BP | Paleofeces | CANID or CERVID | 12 |
| LG 4560.69 | YRK001 | Surrey | UK | Post-Medieval | Paleofeces | HUMAN | 13 |
| AP3-C197S163 | DRL001.A | Derragh | Ireland | Mesolithic | Midden Sediment | – | 14 |
| AP4-A6-2860 | CBA001.A | Cabeço das Amoreiras | Portugal | Mesolithic | Midden Sediment | – | 15 |
| AP5-798-162 | BRF001.A | Binchester Roman Fort | England | Roman | Midden Sediment | – | 16 |
| AP6-LPZ702 | LEI010.A | Leipzig | Germany | 10th–11th century AD | Midden Sediment | – | 17 |
| AP7-6-28353 | ECO004.D | El Collado | Spain | Mesolithic | Pelvic Sediment | – | 18 |
| AP8-CMN-M1 | CMN001.D | Cingle del Mas Nou | Spain | Mesolithic | Pelvic Sediment | – | 19 |
| AP9-17590 | MLP001.A | Molpir | Slovakia | 7th century BC | Pelvic Sediment | – | 20 |

**Note:**

* Metagenomic data were previously published in *Hagan et al. (2020)*.

## Archaeological samples

A total of 20 archaeological samples, originating from 10 sites (Fig. S3) and spanning periods from 7200 BP to the medieval era, were selected for this study. Among these 20 samples, of which 17 are newly sequenced, 13 are paleofeces, 4 are midden sediments, and 3 are sediments obtained from human pelvic bone surfaces (Table 2).

## Sampling

Paleofeces specimens from Mexico were sampled in a dedicated aDNA cleanroom in the Laboratories for Molecular Anthropology and Microbiome Research (LMAMR) at the University of Oklahoma, USA. Specimens from China were sampled in a dedicated aDNA cleanroom at the Max Planck Institute for the Science of Human History (MPI-SHH) in Jena, Germany. All other specimens were first sampled at the Max Planck Institute for Evolutionary Anthropology (MPI-EVA) in Leipzig, Germany before being transferred to the MPI-SHH for further processing. Sampling was performed using a sterile stainless steel spatula or scalpel, followed by homogenization in a mortar and pestle, if necessary. Because the specimens from Xiaosungang, China were very hard and dense, a rotary drill was used to section the coprolite prior to sampling. Where possible, fecal material was sampled from the interior of the specimen rather than the surface. Specimens from Molphir and Leipzig were received suspended in a buffer of trisodium phosphate, glycerol, and formyl following screening for parasite eggs using optical microscopy. For each paleofeces specimen, a total of 50–200 mg was analyzed.

Modern feces were obtained under written informed consent from Boston, USA (WHU) from a long-term (>50 years) type 1 diabetes cohort, and from villages in Burkina Faso (NWHR) as part of broader studies on human gut microbiome biodiversity and health-associated microbial communities. Feces were collected fresh and stored frozen until analysis. A total of 250 mg was analyzed for each fecal specimen.

## DNA extraction

For paleofeces and sediment samples, DNA extractions were performed using a silica spin column protocol (*Dabney et al., 2013*) with minor modifications in dedicated aDNA cleanrooms located at LMAMR (Mexican paleofeces) and the MPI-SHH (all other paleofeces). At LMAMR, the modifications followed those of method D described in *Hagan et al. (2020)*. DNA extractions at the MPI-SHH were similar, but omitted the initial bead-beating step, and a single silica column was used per sample instead of two. Additionally, to reduce centrifugation errors, DNA extractions performed at the MPI-SHH substituted the column apparatus from the High Pure Viral Nucleic Acid Large Volume Kit (Roche, Switzerland) in place of the custom assembled Zymo-reservoirs coupled to MinElute (Qiagen, Hilden, Germany) columns described in *Dabney et al. (2013)*. Samples processed at the MPI-SHH were also partially treated with uracil-DNA-glycosylase (UDG) enzyme to confine DNA damage to the ends of the DNA molecules (*Rohland et al., 2015*).

For modern feces, DNA was extracted from Burkina Faso fecal samples using the AllPrep PowerViral DNA/RNA Qiagen kit at Centre MURAZ Research Institute in Burkina Faso. DNA was extracted from the Boston fecal material using the ZymoBIOMICS DNA Miniprep Kit (D4303) at the Joslin Diabetes Center.

## Library preparation and sequencing

For paleofeces and sediment samples, double-stranded, dual-indexed shotgun Illumina libraries were constructed following (*Meyer & Kircher, 2010*) using either the NEBNext DNA Library Prep Master Set (E6070) kit (*Hagan et al., 2020*; *Mann et al., 2018*) for the

Mexican paleofeces or individually purchased reagents (*Mann et al., 2018*) for all other samples. Following library amplification using Phusion HotStart II (ZSM023, ZSM025, ZSM027, ZSM029), KAPA HiFi Uracil+ (ZSM002, ZSM005, ZSM028), or Agilent Pfu Turbo Cx Hotstart (all other paleofeces) polymerase, the libraries were purified using a Qiagen MinElute PCR Purification kit and quantified using either a BioAnalyzer 2100 with High Sensitivity DNA reagents or an Agilent Tape Station D1000 Screen Tape kit. The Mexican libraries were pooled in equimolar amounts and sequenced on an Illumina HiSeq 2000 using $2 \times 100$ bp paired-end sequencing. All other libraries were pooled in equimolar amounts and sequenced on an Illumina HiSeq 4000 using $2 \times 75$ bp paired-end sequencing.

For modern NWHR feces, double-stranded, dual-indexed shotgun Illumina libraries were constructed in a dedicated modern DNA facility at LMAMR. Briefly, after DNA quantification using a Qubit dsDNA Broad Range Assay Kit, DNA was sheared using a QSonica Q800R in 1.5 mL 4 °C cold water at 50% amplitude for 12 min to aim for a fragment size between 400 and 600 bp. Fragments shorter than 150 bp were removed using Sera-Mag SpeedBeads and a Alpaqua 96S Super Magnet Plate. End-repair and A-tailing was performed using the Kapa HyperPrep EndRepair and A-Tailing Kit, and Illumina sequencing adapters were added. After library quantification, libraries were dual-indexed in an indexing PCR over four replicates, pooled, and purified using the SpeedBeads. Libraries were quantified using the Agilent Fragment Analyzer, pooled in equimolar ratios, and size-selected using the Pippin Prep to a target size range of 400–600 bp. Libraries were sequenced on an Illumina NovaSeq S1 using $2 \times 150$ bp paired-end sequencing at the Oklahoma Medical Research Foundation Next-Generation Sequencing Core facility. Modern WHU libraries were generated using the NEBNext DNA library preparation kit following manufacturer's recommendations, after fragmentation by shearing for a target fragment size of 350 bp. The libraries were then pooled and sequenced by Novogene on a NovaSeq S4 using $2 \times 150$ bp paired-end sequencing.

## Proportion of host DNA in gut microbiome

Because it is standard practice to remove human DNA sequences from metagenomics DNA sequence files before data deposition into public repositories, we were unable to infer the proportion of human DNA in human feces from publicly available data. To overcome this problem, we measured the proportion of human DNA in two newly generated fecal metagenomics datasets from Burkina Faso (NWHR) and Boston, U.S.A. (WHU) (Table S5). To measure the proportion of human DNA in each fecal dataset, we used the Anonymap pipeline (*Borry, 2019a*) to perform a mapping with Bowtie 2 (*Langmead & Salzberg, 2012*) with the parameters `-- very-sensitive -N 1` after adapter cleaning and reads trimming for ambiguous and low-quality bases with a QScore below 20 by AdapterRemoval v2 (*Schubert, Lindgreen & Orlando, 2016*). To preserve the anonymity of the donors, the sequences of mapped reads were then replaced by Ns thus anonymizing the alignment files. We obtained the proportion of host DNA per sample by dividing the number of mapped reads by the total number of reads in the sample. The proportion of

host DNA in dog feces was determined from the published dataset *Coelho et al. (2018)* as described above, but without the anonymization step.

## Visualization and statistical analysis

The statistical analyses were performed in Python v3.7.6 using Scipy v1.4.1, and the figures were generated using Plotnine v0.6.0.

## coproID pipeline

Data were processed using the coproID pipeline v1.0 (Fig. 2) (DOI 10.5281/zenodo.2653757) written using Nextflow (*Di Tommaso et al., 2017*) and made available through nf-core (*Ewels et al., 2019*). Nextflow is a Domain Specific Language designed to ensure reproducibility and scalability for scientific pipelines, and nf-core is a community-developed set of guidelines and tools to promote standardization and maximum usability of Nextflow pipelines. CoproID consists of 5 different steps:

### Preprocessing

*Fastq* sequencing files are given as an input. After quality control analysis with FastQC (*Andrews, 2010*), raw sequencing reads are cleaned from sequencing adapters and trimmed from ambiguous and low-quality bases with a QScore below 20, while reads shorter than 30 base pairs are discarded using AdapterRemoval v2. By default, paired-end reads are merged on overlapping base pairs.

### Mapping

The preprocessed reads are then aligned to each of the target species genomes (source species) by Bowtie2 with the `-- very-sensitive` preset while allowing for a mismatch in the seed search (`-N 1`). When running coproID with the ancient DNA mode (`--adna`), alignments are filtered by PMDtools (*Skoglund et al., 2014*) to only retain reads showing post-mortem damages (PMD). PMDtools default settings are used, with specified library type, and only reads with a PMDScore greater than three are kept.

### Computing host DNA content

Next, filtered alignments are processed in Python using the Pysam library (*Pysam Developers, 2018*). Reads matching above the identity threshold of 0.95 to multiple host genomes are flagged as common reads $reads_{commons}$ whereas reads mapping above the identity threshold to a single host genome are flagged as genome-specific host reads $reads_{spec\ g}$ to each genome $g$. Each source species host DNA is normalized by genome size and gut microbiome host DNA content such as:

$$NormalizedHostDNA(source\ species) = \frac{\Sigma length(reads_{spec\ g})}{genome_{g\ length} \cdot endo_g} \qquad (1)$$

where for each species of genome $g$, $\Sigma length(reads_{spec\ g})$ is the total length of all $reads_{spec\ g}$, $genome_{g\ length}$ is the size of the genome, and $endo_g$ is the host DNA proportion in the species gut microbiome.
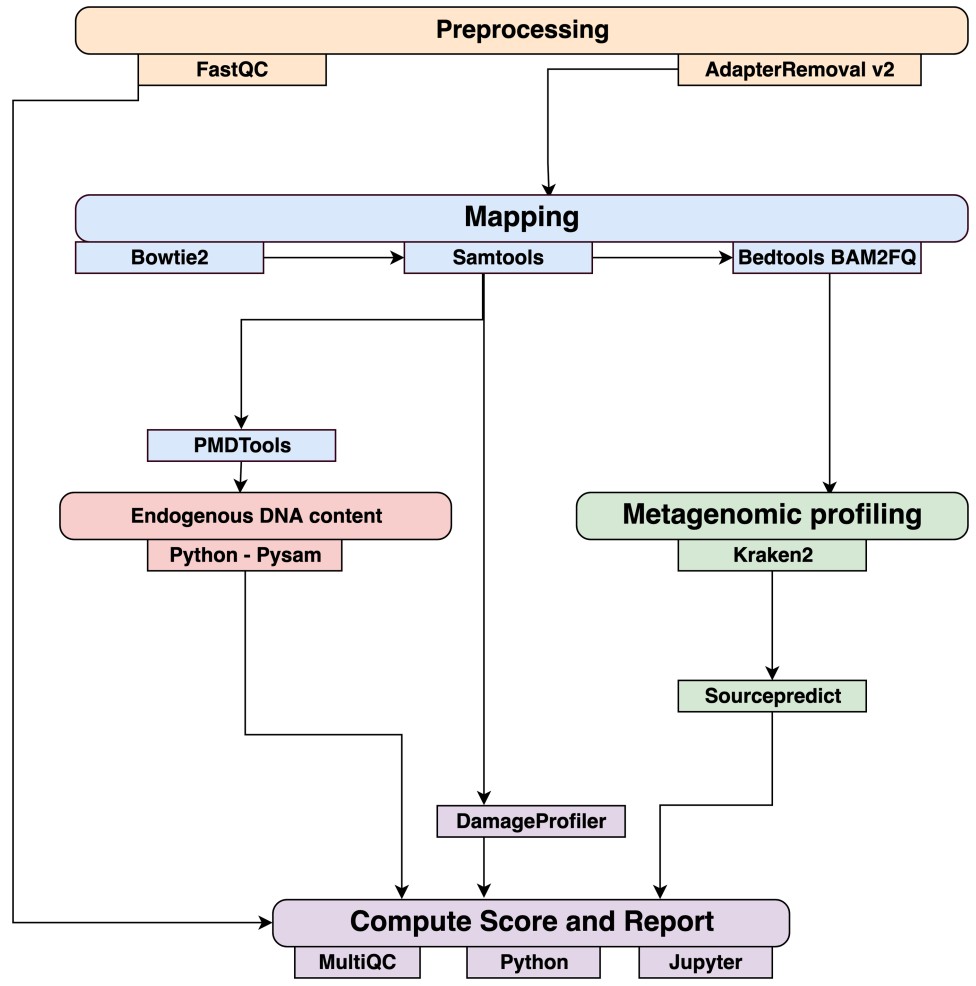

**Figure 2 Workflow schematic of the coproID pipeline.** CoproID consists of five steps: *Preprocessing* (orange), *Mapping* (blue), *Computing host DNA content for each metagenome* (red), *Metagenomic profiling* (green), and *Reporting* (violet). Individual programs (squared boxes) are colored by category (rounded boxes).

Afterwards, an host DNA ratio is computed for each source species such as:

$$NormalizedRatio(source\ species) = \frac{NormalizedHostDNA(source\ species)}{\Sigma NormalizedHost\ DNA(source\ species)} \quad (2)$$

where Σ*NormalizedHost DNA*(*source species*) is the sum of all source species Normalized Host DNA.

### Metagenomic profiling

Adapter clipped and trimmed reads are given as an input to Kraken 2 (*Wood & Salzberg, 2014*). Using the MiniKraken2_v2_8GB database (2019/04/23 version), Kraken 2 performs the taxonomic classification to output a taxon count per sample report file. All samples' taxon counts are pooled together in a taxon counts matrix with samples in columns, and taxons in rows. Next, Sourcepredict (*Borry, 2019b*) is used to predict the source based on each microbiome sample taxon composition. Using dimension reduction and K-Nearest

Neighbors (KNN) machine learning trained with reference modern gut microbiomes samples (Table 1), Sourcepredict estimates a proportion $prop_{microbiome}(source\ species)$ of each potential source species, here Human or Dog, for each sample.

### Reporting

For each filtered alignment file, the DNA damage patterns are estimated with DamageProfiler (*Peltzer & Neukamm, 2019*). The information from the host DNA content and the metagenomic profiling are gathered for each source in each sample such as:

$$proportion(source\ species) = NormalizedRatio(source\ species) \cdot$$
$$prop_{microbiome}(source\ species)$$

Finally, a summary report is generated including the damage plots, a summary table of the coproID metrics, and the embedding of the samples in two dimensions by Sourcepredict. coproID is available on GitHub at the following address: github.com/nf-core/coproid.

## RESULTS

We analyzed 20 archaeological samples with coproID v1.0 to estimate their source using both host DNA and microbiome composition.

### Host DNA in reference gut microbiomes

Before analyzing the archaeological samples, we first tested whether there is a per-species difference in host DNA content in modern reference human and dog feces. With Anonymap, we computed the amount of host DNA in each reference gut microbiome (Table S1). We found that the median percentages of host DNA in NWHR, WHU, and Dog (Fig. 3) are significantly different at $alpha$ = 0.05 (Kruskal–Wallis $H$-test = 117.40, $p$ value < 0.0001). We confirmed that there is a significant difference of median percentages of host DNA between dogs and NWHR, as well as dogs and WHU, with Mann–Whitney $U$ tests (Table 3) and therefore corrected each sample by the mean percentage of gut host DNA found in each species, 1.24% for humans ($\mu_{NWHR}$ = 0.85, $\sigma_{NWHR}$ = 2.33, $\mu_{WHU}$ = 1.67, $\sigma_{WHU}$0.81), and 0.11% for dogs ($\sigma_{dog}$ = 0.16) (Eq. (1); Table S1). This information was used to correct for the amount of host DNA found in paleofeces.

### The effect of PMD filtering on host species prediction

Because aDNA accumulates damage over time (*Briggs et al., 2007*), we could use this characteristic to filter for reads carrying these specific damage patterns using PMDtools, and therefore reduce modern contamination in the dataset. We applied PMD filtering to our archaeological datasets, and for each, compared the predicted host source before and afterwards. The predicted host sources did not change after the DNA damage read filtering, but some became less certain (Fig. 4). Most samples are confidently assigned to one of the two target species, however some samples previously categorized as humans now lie in the uncertainty zone. This suggests that PMDtools filtering lowered the modern

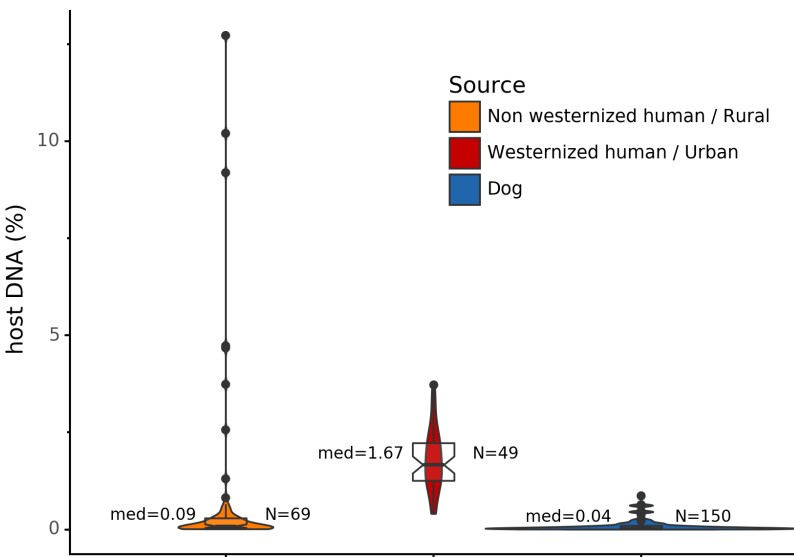

**Figure 3 Gut microbiome host DNA content.** The median percentage of host DNA in the gut microbiome and the number of samples in each group are displayed besides each boxplot.

**Table 3 Statistical comparison of reference gut host DNA content.** Mann–Whitney $U$ test for independent observations. $H_0$: the distributions of both populations are equal.

| Comparison | Mann-Whitney $U$ test | $p$ Value |
|---|---|---|
| Dog vs NWHR | 3327.0 | <0.0001 |
| Dog vs WHU | 41.0 | <0.0001 |
| NWHR vs WHU | 370.0 | <0.0001 |
| Dog vs Human | 3368.0 | <0.0001 |

human contamination which might have originated from sample excavation and manipulation.

The trade-off of PMDtools filtering is that it reduces the assignment power by lowering the number of reads available for host DNA-based source prediction by only keeping PMD-bearing reads. This loss is greater for well-preserved samples, which may have relatively few damaged reads (<15% of total). Ultimately, applying damage filtering can make it more difficult to categorize samples on the sole basis of host DNA content, but it also makes source assignments more reliable by removing modern contamination.

## Source microbiome prediction of reference samples by Sourcepredict

To help resolve ambiguities related to the host aDNA present within a sample, we also investigated gut microbiome composition as an additional line of evidence to better predict paleofeces source. After performing taxonomic classification using Kraken2, we computed a sample pairwise distance matrix from the species counts. With the $t$-SNE dimension reduction method, we embedded this distance matrix in two dimensions to visualize the

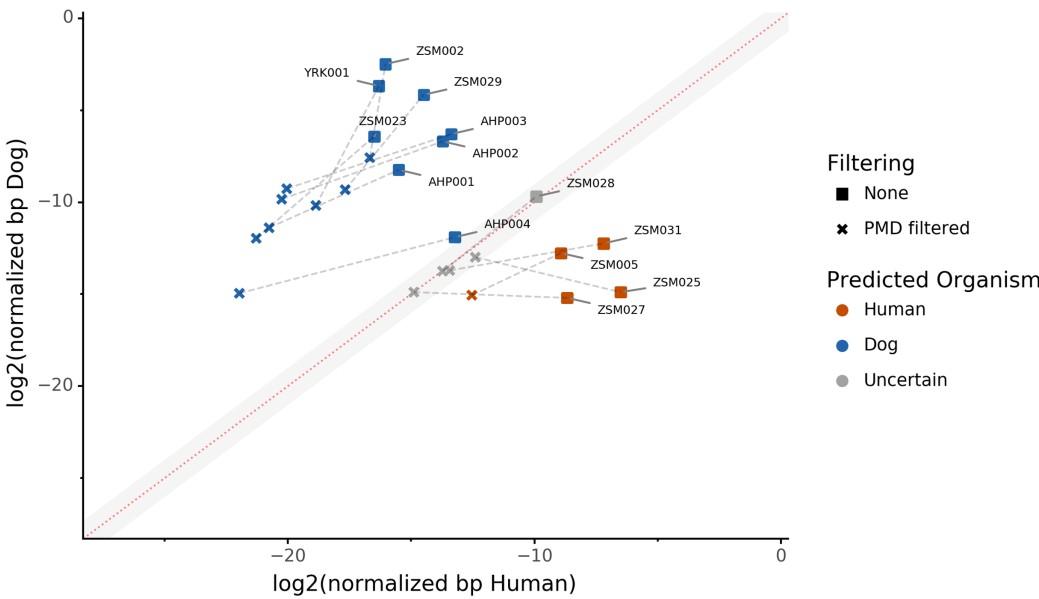

**Figure 4 The effect of filtering for damaged reads using PMD.** The log2 of the human *NormalizedHostDNA* is graphed against the log2 of the dog *NormalizedHostDNA*. Squares represent samples before filtering by PMD, whereas crosses represent samples after filtering by PMD. Dotted lines show the correspondence between samples. The red diagonal line marks the boundary between the two species, and the grey shaded area indicates a zone of species uncertainty (±1log2FC) due to insufficient genetic information.

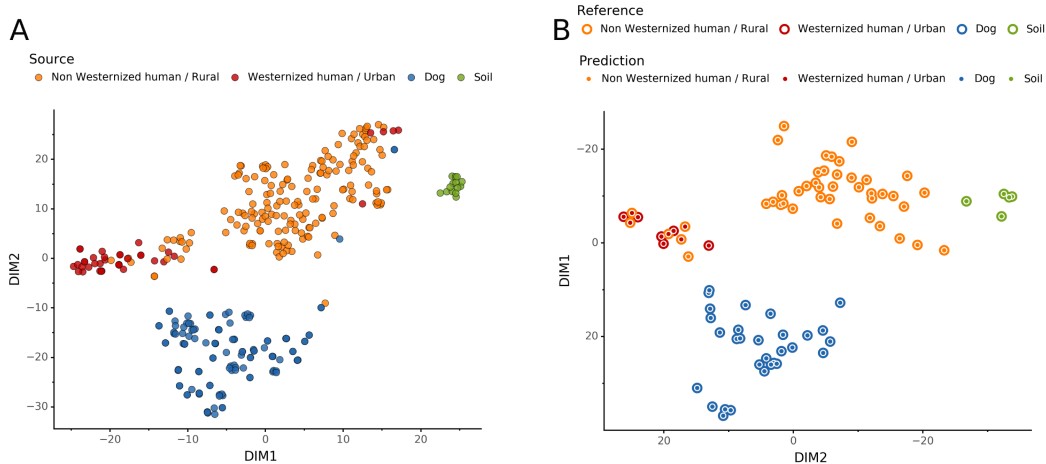

**Figure 5 Embedding of reference modern gut microbiomes.** (A) *t*-SNE embedding of the species composition based on sample pairwise Weighted Unifrac distances for training modern gut microbiomes training samples. Samples are colored by their actual source. (B) *t*-SNE embedding of the species composition based on sample pairwise Weighted Unifrac distances for source prediction of modern test samples. The outer circle color is the actual source of a sample, while the inner circle color is the predicted sample source by Sourcepredict.

sample positions and sources (Fig. 5A). We then used a KNN machine learning classifier on this low dimension embedding to predict the source of gut microbiome samples. This trained KNN model reached a test accuracy of 0.94 on previously unseen data (Fig. 5B).

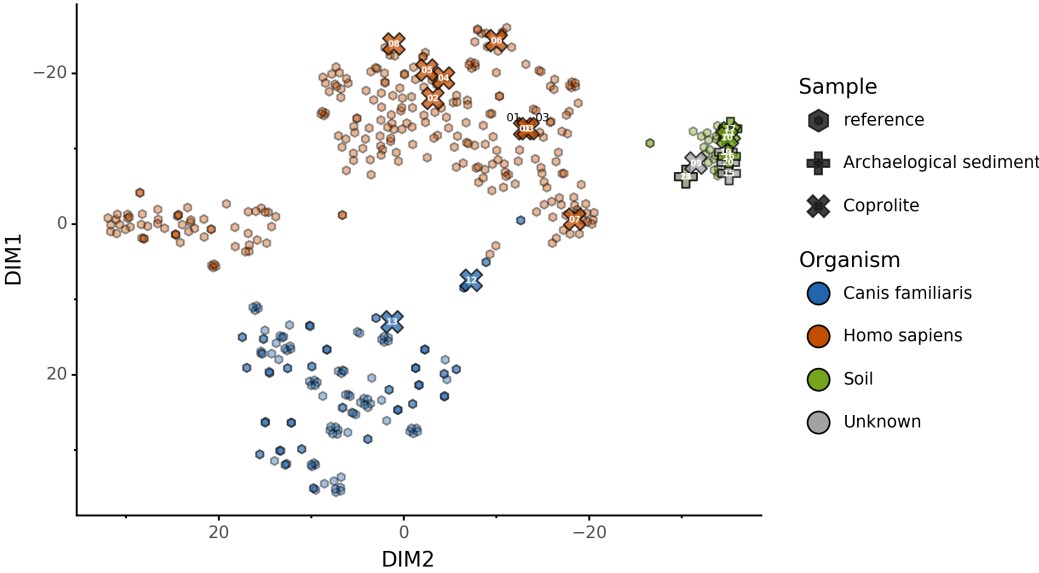

**Figure 6 Prediction of archaeological samples sources and *t*-SNE embedding by Sourcepredict.** *t*-SNE embedding of archaeological (crosses) and modern (hexagons) samples. The color of the modern samples is based on their actual source while the color of the archaeological samples is based on their predicted source by Sourcepredict. Archaeological sample are labelled with their *Plot ID* (Table 2).

## Embedding of archaeological samples by Sourcepredict

We used this trained KNN model to predict the sources of the 20 paleofeces and archaeological sediment samples, after embedding them in a two-dimensional space (Fig. 6). Based on their microbiome composition data, Sourcepredict predicted 2 paleofeces samples as dogs, 8 paleofeces samples as human, 2 paleofeces samples and 4 archaeological sediments as soil, while the rest were predicted as unknown (Table S2).

## coproID prediction

Combining both PMD-filtered host DNA information and microbiome composition, coproID was able to reliably categorize 7 of the 13 paleofeces samples, as 5 human paleofeces and 2 canine paleofeces, whereas all of the non-fecal archaeological sediments were flagged as unknown (Fig. 7). This confirms the original archaeological source hypothesis for five samples (ZSM005, ZSM025, ZSM027, ZSM028, ZSM031) and specifies or rejects the original archaeological source hypothesis for the two others (YRK001, AHP004). The 6 paleofeces samples not reliably identified by coproID have a conflicting source proportion estimation between host DNA and microbiome composition (Fig. 8; Table S3). Specifically, paleofeces AHP001, AHP002 and AHP003 show little predicted gut microbiome preservation, and thus have likely been altered by taphonomic (decomposition) processes. Paleofeces ZSM002, ZSM023 and ZSM029, by contrast, show good evidence of both host and microbiome preservation, but have conflicting source predictions based on host and microbiome evidence. Given that subsistence is associated with gut microbiome composition, this conflict may be related to insufficient gut microbiome datasets available for non-Westernized dog populations (*Hagan et al., 2020*).

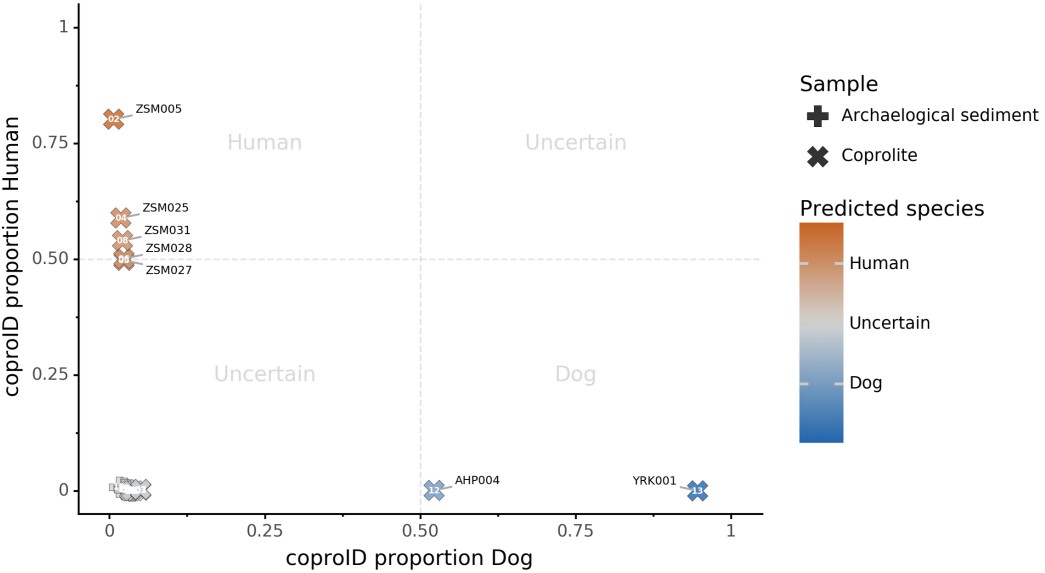

**Figure 7 coproID source prediction.** Predicted human proportion graphed versus predicted canine proportion. Samples are colored by their predicted sources proportions. Samples with a low canine and human proportion are not annotated.

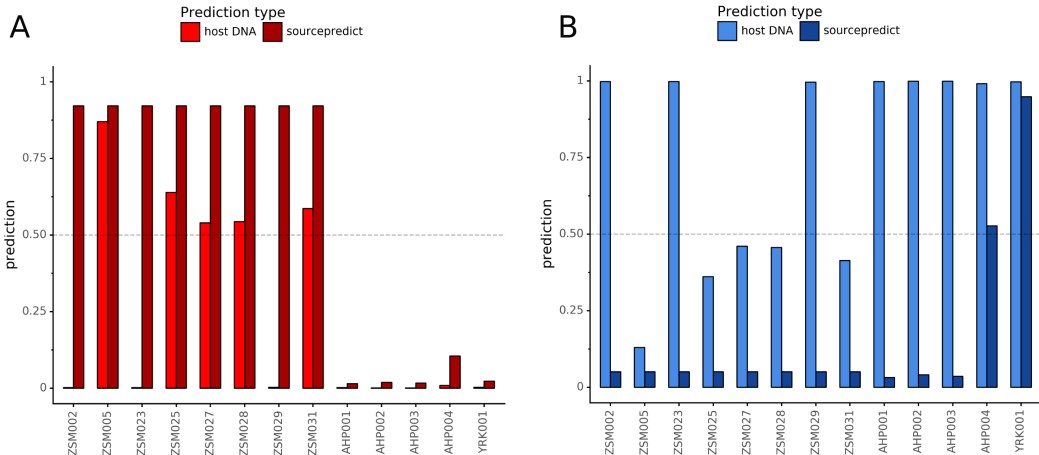

**Figure 8 Host DNA and Sourcepredict source prediction for paleofeces samples.** For human (A) and canine (B). The vertical bar represents the predicted proportion by host DNA (lighter fill) or by Sourcepredict (darker fill). The horizontal dashed line represents the confidence threshold to assign a source to a sample.

## DISCUSSION

Paleofeces are the preserved remains of human or animal feces, and although they typically only preserve under highly particular conditions, they are nevertheless widely reported in the paleontological and archaeological records and include specimens ranging in age from the Paleozoic era (*Dentzien-Dias et al., 2013*) to the last few centuries. Paleofeces can provide unprecedented insights into animal health and diet, parasite biology and evolution, and the changing ecology and evolution of the gut microbiome. However,

because many paleofeces lack distinctive morphological features, determining the host origin of a paleofeces can be a difficult problem (*Poinar et al., 2009*). In particular, distinguishing human and canine paleofeces can be challenging because they are often similar in size and shape, they tend to co-occur at archaeological sites and in midden deposits, and humans and domesticated dogs tend to eat similar diets (*Guiry, 2012*). We developed coproID to aid in identifying the source organism of archaeological paleofeces and coprolites by applying a combined approach relying on both ancient host DNA content and gut microbiome composition.

coproID addresses several shortcomings of previous methods. First, we have included a DNA damage-filtering step that allows for the removal of potentially contaminating modern human DNA, which may otherwise skew host species assignment. We have additionally measured and accounted for significant differences in the mean proportion of host DNA found in dog and human feces, and we also accounted for differences in host genome size between humans and dogs when making quantitative comparisons of host DNA. Then, because animal DNA recovered from paleofeces may contain a mixture of host and dietary DNA, we also utilize gut microbiome compositional data to estimate host source. We show that humans and dogs have distinct gut microbiome compositions, and that their feces can be accurately distinguished from each other and from non-feces using a machine learning classifier after data dimensionality reduction. Taken together, these approaches allow a robust determination of paleofeces and coprolite host source, that takes into account both modern contamination, microbiome composition, and postmortem degradation.

In applying coproID to a set of 20 archaeological samples of known and/or suspected origin, all 7 non-fecal sediment samples were accurately classified as "uncertain" and were grouped with soil by Sourcepredict. For the 13 paleofeces and coprolites under study, 7 exhibited matching host and microbiome source assignments and were confidently classified as either human ($n = 5$) or canine ($n = 2$). Importantly, one of the samples confidently identified as canine was YRK001, a paleofeces that had been recovered from an archaeological chamber pot in the United Kingdom, but which showed an unusual diversity of parasites inconsistent with human feces, and therefore posed issues in host assignation.

For the remaining six unidentified paleofeces, three exhibited poor microbiome preservation and were classified as "uncertain", while the other three were well-preserved but yielded conflicting host DNA and microbiome assignments. These three samples, ZSM002, Z023 and ZSM029, all from prehistoric Mexico, all contain high levels of canine DNA, but have gut microbiome profiles within the range of NWHR humans. Classified as "uncertain", there are two possible explanations for these samples. First, these feces could have originated from a human who consumed a recent meal of canine meat. Dogs were consumed in ancient Mesoamerica (*Clutton-Brock & Hammond, 1994*; *Santley & Rose, 1979*; *Rosenswig, 2007*; *Wing, 1978*), but further research on the expected proportion of dietary DNA in human feces is needed to determine whether this is a plausible explanation for the very high amounts of canine DNA (and negligible amounts of human DNA) observed.

Alternatively, these feces could have originated from a canine whose microbiome composition is shifted relative to that of the reference metagenomes used in our training set. It is now well-established that subsistence mode strongly influences gut microbiome composition in humans (*Obregon-Tito et al., 2015*), with NWHR and WHU human populations largely exhibiting distinct gut microbiome structure (Fig. 5A). To date, no gut microbiome data is available from non-Westernized dogs, and all reference dog metagenome data included as training data for coproID originated from a single study of labrador retrievers and beagles (*Coelho et al., 2018*). Future studies of non-Westernized rural dogs are needed to establish the full range of gut microbial diversity in dogs and to more accurately model dog gut microbiome diversity in the past. Given that all confirmed human paleofeces in this study falls within the NWHR cluster (Fig. 6), we anticipate that our ability to accurately classify dog paleofeces and coprolites as canine (as opposed to "uncertain") will improve with the future addition of non-Westernized rural dog metagenomic data.

In addition to archaeological applications, coproID may also have useful applications in the field of forensic genetic sciences, where it may assist with the identification of human or other feces. As with the investigation of paleofeces, coproID works best when sufficient comparative reference materials or datasets are available. Until a more exhaustive catalog of the human and dog gut microbiome composition is established, not all samples submitted to the coproID analysis will be able to be accurately classified. However, as microbiome reference datasets expand and methods become more standardized in the field, gut microbiome analyses will have increasing applications in the fields of archaeology and forensics (*Hampton-Marcell, Lopez & Gilbert, 2017*).

## CONCLUSIONS

We developed an open-source, documented, tested, scalable, and reproducible method to perform the identification of archaeological paleofeces and coprolite source. By leveraging the information from host DNA and microbiome composition, we were able to identify and/or confirm the source of newly sequenced paleofeces. We demonstrated that coproID can provide useful assistance to archaeologists in identifying authentic paleofeces and inferring their host. Future work on dog gut microbiome diversity, especially among rural, non-Westernized dogs, may help improve the tool's sensitivity even further.

## ACKNOWLEDGEMENTS

We thank David Petts, Zdeněk Tvrdý, Susanne Stegmann-Rajtár, and Zuzana Rajtarova for contributing archaeological samples to this study. We thank the Guildford Museum (Guildford Borough Council Heritage Service) and Catriona Wilson for allowing us to analyze the chamber pot paleofeces sample from Surrey, UK. The sample from Derragh, Ireland was excavated by Discovery Programme, an all-Ireland public center of archaeological research supported by the Heritage Council, during field work in 2003–2005 as part of the Lake Settlement Project. Thanks to the Servei d'Investigació Prehistòrica of València and Museu de la Valltorta of Catelló for access to material.

### Funding

This work was supported by the US National Institutes of Health R01GM089886 (to Christina Warinner and Cecil Lewis), the Deutsche Forschungsgemeinschaft EXC 2051 #390713860 (to Christina Warinner), and the Max Planck Society. The funders had no role in study design, data collection and analysis, decision to publish, or preparation of the manuscript.

### Grant Disclosures

The following grant information was disclosed by the authors:
US National Institutes of Health R01GM089886.
Deutsche Forschungsgemeinschaft: EXC 2051 #390713860.
Max Planck Society.

### Competing Interests

The authors declare that they have no competing interests.

### Author Contributions

- Maxime Borry conceived and designed the experiments, analyzed the data, prepared figures and/or tables, authored or reviewed drafts of the paper, and approved the final draft.
- Bryan Cordova performed the experiments, authored or reviewed drafts of the paper, and approved the final draft.
- Angela Perri performed the experiments, authored or reviewed drafts of the paper, and approved the final draft.
- Marsha Wibowo performed the experiments, analyzed the data, authored or reviewed drafts of the paper, and approved the final draft.
- Tanvi Prasad Honap performed the experiments, analyzed the data, authored or reviewed drafts of the paper, and approved the final draft.
- Jada Ko performed the experiments, authored or reviewed drafts of the paper, and approved the final draft.
- Jie Yu performed the experiments, authored or reviewed drafts of the paper, and approved the final draft.
- Kate Britton performed the experiments, authored or reviewed drafts of the paper, and approved the final draft.
- Linus Girdland-Flink performed the experiments, authored or reviewed drafts of the paper, and approved the final draft.
- Robert C. Power performed the experiments, authored or reviewed drafts of the paper, and approved the final draft.
- Ingelise Stuijts performed the experiments, authored or reviewed drafts of the paper, and approved the final draft.

# PeerJ

- Domingo C. Salazar-García performed the experiments, authored or reviewed drafts of the paper, and approved the final draft.
- Courtney Hofman performed the experiments, authored or reviewed drafts of the paper, and approved the final draft.
- Richard Hagan performed the experiments, authored or reviewed drafts of the paper, and approved the final draft.
- Thérèse Samdapawindé Kagoné performed the experiments, authored or reviewed drafts of the paper, and approved the final draft.
- Nicolas Meda performed the experiments, authored or reviewed drafts of the paper, and approved the final draft.
- Helene Carabin performed the experiments, authored or reviewed drafts of the paper, and approved the final draft.
- David Jacobson performed the experiments, authored or reviewed drafts of the paper, and approved the final draft.
- Karl Reinhard performed the experiments, authored or reviewed drafts of the paper, and approved the final draft.
- Cecil Lewis conceived and designed the experiments, authored or reviewed drafts of the paper, and approved the final draft.
- Aleksandar Kostic conceived and designed the experiments, authored or reviewed drafts of the paper, and approved the final draft.
- Choongwon Jeong analyzed the data, authored or reviewed drafts of the paper, and approved the final draft.
- Alexander Herbig conceived and designed the experiments, authored or reviewed drafts of the paper, and approved the final draft.
- Alexander Hübner conceived and designed the experiments, authored or reviewed drafts of the paper, and approved the final draft.
- Christina Warinner conceived and designed the experiments, analyzed the data, prepared figures and/or tables, authored or reviewed drafts of the paper, and approved the final draft.

## Human Ethics

The following information was supplied relating to ethical approvals (i.e., approving body and any reference numbers):

The Joslin Diabetes Center granted ethical approval to sample the WHU individuals (Study No. 2017-25). The Centre MURAZ Research Institute granted ethical approval to sample the NWHR individuals (31/2016/CE-CM).

## DNA Deposition

The following information was supplied regarding the deposition of DNA sequences:

Genetic data are available in the European Nucleotide Archive (ENA): PRJEB33577 and PRJEB35362.

## Data Availability
The code for the analysis is available at GitHub: https://github.com/maxibor/coproid-article and Zenodo:

Maxime Borry, Alexander Peltzer, & James A. Fellows Yates. (2019, April 29). nf-core/coproid: coproID v1.0 - Dioptre Walrus (Version 1.0). Zenodo. DOI 10.5281/zenodo.2653757.

## Supplemental Information
Supplemental information for this article can be found online at http://dx.doi.org/10.7717/peerj.9001#supplemental-information.

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
