# Peer review of "CoproID predicts the source of coprolites and paleofeces using microbiome composition and host DNA content"

_PeerJ, doi:10.7717/peerj.9001_

## Round 0.1 · original submission · Minor Revisions

Besides the Reviewers comments, I would very much appreciate if you could add some reflection on the interaction with Forensic Genetics perspective and methods.

Reviewer 1 ·

Basic reporting

In the work, the authors present a bioinformatics method to infer and authenticate the host source of paleofeces from shotgun metagenomic DNA sequencing data.
At first look, this is a very good manuscript. At the same time, after its reading it is clear that the authors where first who implemented the identification system.
The authors present a reproducible method using both host and microbial DNA to infer the host source of fecal material of archaeological contexts. This is extremely interesting in ancestral studies. So also about migrations and interactions between peoples and cultures in different geographical sites.
The introduction is clear and shows the context and general background of the investigation. Literature agrees with the characteristics of the work. References are relevant.
In the document the figures are of high quality and explanatory. (Fig 1 A, B, C) although the size is not visible. I suggest putting a unit of measure in the images.

Experimental design

I think that this work is an original research and very high impact. The numbers of samples is adequate and research aims well defined.
I suggest incorporating a map that indicates the sampling points. Add georeference of each point.
Methods described are very details and high probability replicated.
Line 149 DNA extraction: It would be interesting to know if the DNA of paleofeces was quantified , how its amplification is verified. DNA extracted from feces and soil is characterized by having a large interfering amount.
In Figure 2. Workflow schematic of the coproID pipeline. Is it the same reported by Maxime Borry (2018). CoproID: Coprolite Identification. DOI: 10.5281/zenodo.1458163 ?
Line 263 aDNA ancient DNA?

Validity of the findings

no comments

Additional comments

The article is a very complete and complex work. It is a contribution to the knowledge of the ancestral interactions of man and animals. It presents a high impact system for the analysis of information from difficult to treat materials (feces, sediments). The methodology is detailed and well defined. I did not see obvious speculation in the writing. I think it should be accepted for publication.

Reviewer 2 ·

Basic reporting

The reporting of article is clear and unambiguous and professional English used throughout. Sufficient background data provided and data is shared appropriately.

Experimental design

no comment

Validity of the findings

no comment

Additional comments

The current research provides open-source, documented, tested, scalable, and reproducible method to perform the identification of archaeological paleofeces and coprolite source. The MS is written according to the guidelines and research questions are well defined, relevant & meaningful. Following are few comments to improve article
Please add citation line 84 "Even well-preserved fecal material degrades over time, changing in size, shape, and color"
Follow format for citation line 104
line 153 "protocol D described" protocol D?????
There are no details of statistical analysis performed. Statistical analysis section should be included at the end of Materials and Methods section and details of analysis performed should be included viz., Kruskal- Wallis H-test and Mann- Whitney U test
line 318 coproID??

---

## Round 0.2 · accepted · Accept

All the suggestions have been followed and criticisms removed.

Reviewer 1 ·

Basic reporting

.

Experimental design

.

Validity of the findings

.